# Painful Experiences in Social Contexts Facilitate Sensitivity to Emotional Signals of Pain from Conspecifics in Laboratory Rats

**DOI:** 10.3390/ani14091280

**Published:** 2024-04-24

**Authors:** Satoshi F. Nakashima, Masatoshi Ukezono, Yuji Takano

**Affiliations:** 1School of Psychological Sciences, University of Human Environments, Matsuyama 790-0825, Japan; y-takano@uhe.ac.jp; 2Department of Developmental Disorders, National Institute of Mental Health, National Center of Neurology and Psychiatry, Tokyo 187-8551, Japan; ukezono@ncnp.go.jp

**Keywords:** emotional signal, pain experience, social referencing

## Abstract

**Simple Summary:**

Receiving emotional signals from conspecifics is crucial for the survival of group-living mammals, such as rodents. It is unclear whether the received visual signal of pain from conspecifics is used for the subsequent behavior of rats; however, previous studies have revealed that rats show emotional expression of pain and differentiate that emotional signal from others. In this study, we manipulated the experience of pain or the presence of other conspecifics to influence the subsequent behavior of rats (i.e., approaching a visual emotional signal of pain or neutral). From these results, it is possible that rats use information received from other conspecifics during manipulation for subsequent behavior as social referencing.

**Abstract:**

Previous studies demonstrated that laboratory rats could visually receive emotional pain signals from conspecifics through pictorial stimuli. The present study examined whether a prior painful emotional experience of the receiver influenced the sensitivity of emotional expression recognition in laboratory rats. The experiment comprised four phases: the baseline preference test, pain manipulation test, post-manipulation preference test, and state anxiety test. In the baseline phase, the rats explored an apparatus comprising two boxes to which pictures of pain or neutral expressions of other conspecifics were attached. In the pain manipulation phase, each rat was allocated to one of three conditions: foot shock alone (pained-alone; PA), foot shock with other unfamiliar conspecifics (pained-with-other; PWO), or no foot shock (control). In the post-manipulation phase, the animals explored the apparatus in the same manner as they did in the baseline phase. Finally, an open-field test was used to measure state anxiety. These findings indicate that rats in the PWO group stayed longer per entry in a box with photographs depicting a neutral disposition than in a box with photographs depicting pain after manipulation. The results of the open-field test showed no significant differences between the groups, suggesting that the increased sensitivity to pain expression in other individuals due to pain experiences in social settings was not due to increased primary state anxiety. Furthermore, the results indicate that rats may use a combination of self-painful experiences and the states of other conspecifics to process the emotional signal of pain from other conspecifics. In addition, changes in the responses of rats to facial expressions in accordance with social experience suggest that the expression function of rats is not only used for emotional expressions but also for communication.

## 1. Introduction

Studies in the field of psychology have long investigated facial expressions, such as joy and fear, using human subjects. However, in recent years, scientific examination of facial expressions has progressed to non-primate mammals, such as dogs [1], cats [2], horses [3], sheep [4], pigs [5], rabbits [6], mice [7], and rats [8]. The quantification of facial expressions in various animal species has become possible, and research on the recognition of facial expressions has advanced [9,10,11]. Additionally, Nakashima et al. showed that rats could visually discriminate between painful and neutral expressions in other conspecifics [12].

Thus, a new question emerges as to whether the facial expressions of various animals are similar to those of humans. Compared with humans, animals have different musculature, which undoubtedly renders their facial expressions different from those of humans. Therefore, a research approach that quantifies expression as a function rather than as a form is required to verify this issue. Waller and Micheletta proposed a theory that defines a facial expression as a signal emitted by an expressor that enhances its adaptation by changing the behavior of the receiver of such a signal [13]. Based on this theory, observing behavioral changes in the receiver is the first step in deciding what signal an animal sends out, similar to human expressions.

Therefore, the present study considered the facial expressions of rodents as a function of communication. To the best of our knowledge, experimental validation of the function of facial expressions in rodents within social contexts is lacking. This study aimed to demonstrate that if facial expressions serve a social function in rats, then the responses of the rats will change depending on the social context in which they witness another rat’s expression. The context used in the experiment was socioemotional with pain expression, which has advanced the study of discrimination in rats. Examining whether the socioemotional context influences the behavior of rats receiving emotional signals from other conspecifics is valuable for exploring the ecology of rats. In addition, it potentially illustrates the general mechanisms of signaling behavior, including those in humans. Increasing evidence has shown that the perception of human emotion varies with socioemotional contexts, such as background information [14,15], body posture [16], environment [17], prior experience [18], and emotional state [19,20,21]. Therefore, evidence of such an effect of socioemotional information on rat responses to emotional signals may indicate a general signaling mechanism, regardless of the species. Therefore, this study examined whether rats perceive signals with socioemotional meaning through visual information (pictures of emotional expressions) by manipulating the socioemotional context. To examine the effect of socioemotional context, the researchers manipulated the experience of rats before observing pictures of a conspecific expressing pain or a neutral disposition. This study comprised four phases: a baseline test, context manipulation, a post-test, and an individual difference examination. In the baseline component, we followed the process of Nakashima et al., where rats in all conditions simply explored an apparatus made up of two boxes to which a picture of either a neutral or painful expression was attached [12]. The reason for selecting still photographs of rats as a stimulus presentation was to strictly control for extraneous variables, such as point-of-view and emotion-unrelated movements, in addition to emotional signals. For context manipulation, the rats were categorized into three groups: pain-alone (PA), pain-with-other (PWO), and control. A clear plastic wall was used to divide the square box into shock and observation rooms. Animals in the PA group were placed alone in a shock room with a grid floor and received an electric shock to their feet. Rats in the PWO group were placed in a shock room and electrically shocked in the same manner as rats in the PA group. Another rat was concurrently placed in an observation room. If facial expressions serve a social function in rats, they would be affected by manipulation in the present experiment. Finally, an open-field test was conducted because the context used may have led to individual differences in the impact of each individual on state anxiety.

The first hypothesis was that the mere experience of pain contributes to elucidating the pain situation of other conspecifics through visual information (pain experience hypothesis). In previous studies, rats with previous experience of foot shock displayed compensatory freezing when witnessing other individuals being foot-shocked [22]. Accordingly, the PA and PWO groups were more likely to avoid pain stimuli after manipulation, which supports the pain experience hypothesis. If this is true, then the PA and PWO groups will demonstrate a tendency to avoid the picture of the pain of other conspecifics to a greater extent than the control group.

The second hypothesis was that the social reference of the rats’ pain experiences would influence their understanding of the pain situation of other conspecifics through visual information (social-referencing hypothesis). If this were true, then the presence of other conspecifics without pain expression would facilitate the understanding of the rats’ pain state and the difference between the pain and neutral states. Therefore, rats in the PWO group may discriminate between the painful and neutral states of other conspecifics to a greater extent than the other groups.

If neither hypothesis is supported and the behavior remains the same in the pre- and post-tests, then the facial expressions of rats must be fundamentally different from human expressions and function. If either hypothesis is supported, then the groundwork for the discussion of the functional similarity of facial expressions between humans and rats will have been laid, because pictures of pain show that other individuals alter their behavior. In addition, this study investigated the behavioral changes that occurred during the post-test by observing whether such differences were due to variations in transient state anxiety due to prior foot shock.

## 2. Methods

### 2.1. Animals and Housing

The study involved 60 naïve male Long–Evans rats (Japan SLC, Inc., Shizuoka, Japan) weighing 240–320 g (8–10 weeks old) at the time of the experiment. We used 20 animals per condition, based on the sample size of a previous study [12]. The animals were housed in pairs throughout the acclimation and experimental periods. Each pair was provided approximately 36 g of standard laboratory chow (CE-2, CLEA Japan, Inc., Tokyo, Japan) per day with free access to tap water in the home cage. The temperature, humidity, and light/dark cycle were maintained at 23 ± 1 °C, at 50 ± 5%, and from 08:00 to 20:00 within a rearing system (EBAC-L, CLEA Japan, Inc., Tokyo, Japan), respectively. During the entire experimental period, we assessed the wellness of animals during everyday feeding time to exclude animals from the experiment if they showed poor health; however, all animals were in good health. All procedures were performed in accordance with the guidelines for animal research of the NTT Communication Science Laboratories, based on the Guidelines for Proper Conduct of Animal Experiments [23]. The experimental protocols were approved by the Ethical Review Board of NTT Communication Science Laboratories (no. H27-012).

### 2.2. Design

In this study, a 3 (context: PWO, PA, control) × 2 (phase: pre-/post-pain) × 3 (emotional expression: pain, neutral) mixed model experimental design was used. The first factor was the between-subjects factor. The second and third factors were within-subject factors (i.e., repeated measures).

### 2.3. Pictorial Stimuli

Original photographs of neutral and pained expressions were used in accordance with a previous study [12] (Figure 1B). Photographs were captured using a high-speed camera (FL3-U3-13E4C-C, Point Gray) while the rats were placed inside an operant test chamber. Photographs of neutral expressions were captured after acclimatizing the rats to the environment without the presence of any painful stimuli. In contrast, photographs of the pained expressions were captured while the rats received foot shock. Please refer to Nakashima et al. for details on stimulus generation [12].

### 2.4. Apparatus

#### 2.4.1. Preference Test Box

The preference test was performed using an apparatus to assess conditioned place preference, as described by Nakashima et al. [12]. The apparatus comprised three compartments: two white-sided compartments and a gray central zone. The entrances to each compartment were connected to a central zone. Photographs of three model rats expressing pain were placed on each wall on one side of the compartment, and neutral-expression photographs were placed on the other side.

#### 2.4.2. Apparatus for Manipulation of Context

The apparatus used to manipulate painful experiences consisted of a foot-shock room and an observation room. Both compartments were rectangular operant test chambers (30 cm deep, 30 cm wide, and 63 cm high) with clear sliding plastic walls. The plastic walls were open on one side of each chamber and connected to each other. A plastic wall (35 cm × 60 cm; width × height) with a narrow slit (2 cm) and numerous small holes was inserted between the chambers. Therefore, the animals could not move back and forth between the compartments, although they could see and lightly touch the animals in the other compartments with their noses.

### 2.5. Procedure

The experiment primarily comprised four phases: the baseline preference test, pain experience manipulation, the post-manipulation preference, and the open-field test (Figure 1A). The procedure for each condition was the same, except for pain experience manipulation between conditions. The post-manipulation test was conducted one day after manipulation. The interval between the baseline test and manipulation was more than two weeks to eliminate the effects of the baseline test on the post-manipulation test. An open-field test was conducted the following day to measure state anxiety. The behavior of all animals was recorded, and all outcomes of each procedure (the baseline test, manipulation, post-test, and open-field test) were counted in a blinded manner (The experiment was not blinded because the first author conducted all experimental procedures. Analysis of the data was conducted by the first author, so that process was also not blinded). The rooms used for breeding, pain experience manipulation, and the baseline/post-test were separated and shielded from each other. Each experimental procedure was conducted between 2:00 p.m. and 6:00 p.m., when the activity of the animals was high in the rearing system.

### 2.6. Preference Test

As in a previous study, each rat was placed in the center zone and left to explore the apparatus for 10 min in the baseline and post-manipulation test phases. The total time spent and frequency of entry into each compartment were recorded.

### 2.7. Pain Experience Manipulation

Twenty rats were randomly assigned to each of the three conditions, namely PA, PWO, and control, using a random function in Excel (i.e., we created a random number tied to each animal, changed the sequential order of the animals from the original ID based on a random number, and allocated them to each condition). The animals in the PA group were placed alone in a shock room with a grid floor and received an electrical shock to their feet using a shock generator (O’HARA and CO., LTD, Tokyo, Japan). The intensity was 0.7 mA for 0.5 s. The shock was delivered once per minute five times, accompanied by a beeping sound for 8 s. For the parameters of the intensity and the number of foot shocks, we set bare minimum values from the view of the 3Rs by referring to those used in previous studies on fear conditioning in rats [24,25]. The beeping sound was used as a conditioned stimulus to create a strong fear association between electrical shock and the beeping sound, rather than making animals have an association between electric shock and other environmental situations or making animals have a vague sense of fear in the environment. The animals in the PWO group were placed in a shock room and electrically shocked, similar to those in the PA group. Another rat was placed in an observation room with the same grid floor. However, this room was not connected to an electrical supply, and the observer rats were free from shock. The observer rats were 20 novel individuals that were not included in any conditions of this experiment and were not cagemates of the experimental animals in the PWO group. The animals in the control group were placed in the shock room and experienced the beeping sound in the same manner as the other groups but were never given an electrical shock or an observer rat.

The amount of activity during manipulation was measured by dividing the bottom of the device into equal sections on a monitor and measuring the frequency at which the animal crossed the boundary with or without a beeping sound.

### 2.8. Open-Field Test

Testing was conducted in a 70 × 70 × 40 cm (width × depth × height) square-shaped open field constructed from lumber. The luminance of the central part of the open field was approximately 29 lx, whereas that of the periphery was 16–17 lx. The locomotor activity of all the rats was measured for 10 min. For the analysis, the entire open field was manually divided into nine square regions of equal size, with one central region and eight peripheral regions (Figure 2). The frequency of movement in each region was used to denote the level of anxiety (i.e., more excursions denoted less anxiety). In addition, the number of entries into the central part of the open field was used as an index of the anxiety level (i.e., more entries denoted less anxiety).

## 3. Results

Data from one rat in the PWO condition were excluded due to treatment errors during the procedure. Therefore, the data from 59 rats were used in this study. Initially, to confirm whether the results of our previous study could be replicated, a two-way mixed factorial ANOVA (3 (context) × 2 (emotional expression)) was conducted on the average length of stay in the pre-pain phase. Our results showed a significant effect of emotional expression (*F*(1, 56) = 6.20, *p* = 0.02, *η_p_*^2^ = 0.10) but no main effect of context (*F*(2, 56) = 0.65, *p* = 0.53, *η_p_*^2^ = 0.02) and interaction (*F*(2, 56) = 0.19, *p* = 0.83, *η_p_*^2^ = 0.007). The average length of stay at the location of the neutral disposition picture (*M* = 256.44, *SD* = 44.61) was longer than that of the picture depicting pain (*M* = 229.74, *SD* = 41.02). Therefore, the results of our previous study were replicated in the present study. In addition, the results indicated no difference in the tendency of preference for emotional expression between the groups before context manipulation.

### 3.1. Manipulation

The experimental manipulation was performed after the pre-test. To verify adequate manipulation under each condition, the activity level of the animals during the manipulation was measured. The activity level of each animal was calculated by dividing the in-beep phase by the silent phase (Figure 3). To compare the activity levels of the three groups during the in-beep and silent phases, a one-way between-subjects ANOVA was performed for each score. Our results pointed to a significant difference among conditions during the in-beep sound phase (*F*(2, 56) = 7.80, *p* = 0.01, *η_p_*^2^ = 0.22). Multiple comparisons using Ryan’s method indicated that the activity of animals in the PWO group was lower than those in the PA (*t*(56) = 3.62, *p* < 0.001, *d* = 1.24) and control (*t*(56) = 3.21, *p* = 0.002, *d* = 1.14) groups, whereas no difference was noted between the PA and control groups (*t*(56) = 0.41, *p =* 0.68, *d* = 0.12.). Furthermore, the results illustrated a significant difference among conditions during the silent phase (*F*(2, 56) = 9.23, *p* < 0.001, *η_p_*^2^ = 0.25). Subsequent multiple comparisons using Ryan’s method highlighted that the activity of animals in the PWO and PA groups was lower than that of the control (*t*(56) = 4.28, *p* < 0.001, *d* = 1.28; *t*(56) = 2.47, *p* = 0.016, *d* = 0.92), whereas no difference was observed between the PWO and PA groups (*t*(56) = 1.83, *p* = 0.07, *d* = 0.54).

### 3.2. Post-Pain Phase: Examining the Effects of Manipulation

#### Total Length of Stay

We conducted a three-way mixed factorial ANOVA (3 (context) × 2 (pre-/post-pain) × 2 (emotional expression)) test on the average time (Figure 4). Our results showed a significant main effect of pre-/post-pain (*F*(1, 56) = 14.32, *p* < 0.001, *η_p_*^2^ = 0.20)) and emotional expression (*F*(1, 56) = 13.36, *p* < 0.001, *η_p_*^2^ = 0.19). The average length of stay in the pre-test (*M* = 243.09, *SD* = 44.72) was longer than that in post-test (*M* = 230.17, *SD* = 47.53). The average length of stay at the location of the neutral disposition picture (*M* = 250.74, *SD* = 46.46) was longer than that of the picture depicting pain (*M* = 222.52, *SD* = 46.46). The interaction between context and pre-/post-pain was marginally significant (*F*(1, 56) = 2.56, *p* = 0.08, *η_p_*^2^ = 0.08). A subsequent analysis was conducted and demonstrated that the significant simple main effect of context during the post-test (*F*(2, 112) = 4.22, *p* = 0.02, *η_p_*^2^ = 0.07). A multiple comparisons using Ryan’s method indicated that the length of stay to both boxes in the PA condition was lower than that in the PWO (*t*(112) = 2.78, *p* = 0.006, *d* = 0.35) and control (*t*(112) = 2.09, *p* = 0.04, *d* = 0.34) groups, whereas no difference was noted between the PWO and control groups (*t*(112) = 0.72, *p* = 0.47, *d* = 0.09). However, no differences were observed between the conditions before the manipulation (*F*(2, 112) = 0.29, *p* = 0.75, *η_p_*^2^ = 0.005). Finally, the other main effects and interactions were not significant (all *ps* > 0.10).

### 3.3. Frequency of Entry

We further conducted a three-way mixed factorial ANOVA (3 (context) × 2 (pre-/post-pain) × 2 (emotional expression)) on the frequency of entry (Figure 5). The results demonstrated a significant main effect of context (*F*(2, 56) = 7.48, *p* = 0.001, *η_p_*^2^ = 0.21). A multiple comparison using Ryan’s method indicated that the frequency of entry into both boxes in the PWO condition was lower than that in the PA (*t*(56) = 3.54, *p* < 0.001, d = 0.74) and control (*t*(56) = 3.08, *p* = 0.003, *d* = 0.89) groups, whereas no difference was noted between the PA and control groups (*t*(56) = 0.47, *p =* 0.64, *d* = 0.11). However, the context × pre-/post-pain interaction was also significant (*F*(2, 56) = 3.84, *p* = 0.03, *η_p_*^2^ = 0.12). Therefore, a subsequent analysis was conducted and demonstrated the significant simple main effect of context during the post-test (*F*(2, 112) = 11.14, *p* < 0.001, *η_p_*^2^ = 0.17). A multiple comparisons using Ryan’s method indicated that the frequency of entry into both boxes in the PWO condition was less than that in the PA (*t*(112) = 3.49, *p* < 0.001, *d* = 1.03) and control (*t*(112) = 4.47, *p* < 0.001, *d* = 1.38) conditions after manipulation, whereas no difference was noted between the PA and control groups (*t*(112) = 1.00, *p* = 0.32, *d* = 0.29). However, no differences were observed between the conditions before the manipulation (*F*(2, 113) = 1.30, *p* = 0.28, *η_p_*^2^ = 0.02).

In addition, the main effect of emotional expression was significant (*F*(1, 56) = 6.96, *p* = 0.01, *η_p_*^2^ = 0.11). The frequency of entry into the box depicting pain (*M* = 9.82, *SD* = 2.48) was lower than the neutral disposition (*M* =10.17, *SD* = 2.53). The other main effects and interactions were non-significant (all *ps* > 0.10).

### 3.4. Length of Stay per Entry

Based on the results of the frequency of entry, differences were noted between the PWO and other conditions, which implied that freezing and moving behaviors varied with the conditions. In other words, the ratio of the frequency of entry should be considered when analyzing the length of stay. Therefore, the index “length of stay per entry” was calculated by dividing the average total length of stay by the frequency of entry.

A three-way mixed factorial ANOVA (3 (context) × 2 (pre-/post-pain) × 2 (emotional expression)) was conducted on the length of stay per entry (Figure 6). A significant main effect was observed in terms of context (*F*(2, 56) = 7.43, *p* < 0.01, *η_p_*^2^ = 0.21), emotional expression (*F*(1, 56) = 5.37, *p* < 0.05, *η_p_*^2^ = 0.09), and the context × pre-/post-pain interaction (*F*(2, 56) = 4.24, *p* < 0.05, *η_p_*^2^ = 0.13). However, the context × pre-/post-pain × emotional expression interaction was also significant (*F*(2, 56) = 3.21, *p* < 0.05, *η_p_*^2^ = 0.10). Subsequent analyses revealed that the pre-/post-pain × emotional expression interaction was significant in the PWO condition (*F*(1, 56) = 6.58, *p* < 0.05, *η_p_*^2^ = 0.11),but not in the PA (*F*(1, 56) = 0.81, *p* = 0.37, *η_p_*^2^ = 0.01) or control (*F*(1, 56) = 0.003, *p* = 0.96, *η_p_*^2^ < 0.01) conditions. In the PWO condition, the length of stay per entry at the location of the neutral disposition picture in the post-pain phase was longer than that in the pre-pain phase (*F*(1, 112) = 9.92, *p* < 0.01, *η_p_*^2^ = 0.08). However, the length of stay per entry at the location of the picture depicting pain did not differ between the pre- and post-pain phases (*F*(1, 112) = 0.11, *p* = 0.74, *η_p_*^2^ = 0.001). In addition, animals in the PWO group stayed longer at the location of the neutral picture than at that of the picture depicting pain in the post-pain phase (*F*(1, 112) = 13.83, *p* < 0.001, *η_p_*^2^ = 0.11) but not in the pre-pain phase (*F*(1, 112) = 0.72, *p* = 0.40, *η_p_*^2^ = 0.006).

### 3.5. Open-Field Test

Finally, an open-field test was conducted to measure anxiety state after manipulation. Data from two rats in the PWO condition were not available because of missing data items. Therefore, the data from 57 rats were used in this study. Data were analyzed for the mean proportion of frequency of movement among the nine regions in the open field as an index of the anxiety levels of animals under the three conditions (Figure 7A). As an effect of one-way ANOVA between subjects on the anxiety level score, differences reached marginal significance among conditions (*F*(2, 54) = 2.95, *p* = 0.06, *η_p_*^2^ = 0.10), whereas subsequent multiple comparisons did not show any significant difference among conditions (*n.s.*). Additionally, we analyzed the mean proportion of entries into the central parts of the open field as another index of anxiety (Figure 7B). A one-way between-subjects ANOVA of the index did not show a significant difference between the conditions (*F*(2, 54) = 1.04, *p* = 0.36, *η_p_*^2^ = 0.04).

## 4. Discussion

In support of the social referencing hypothesis, the rats in the PWO condition stayed longer per entry at the neutral picture than at the picture depicting pain after pain manipulation. However, the length of stay in the PA and control groups did not differ between picture locations after manipulation. Thus, rats may have adapted their sensitivity to emotional signals of pain from other conspecifics by socially referencing prior pain experiences. As a result of the increased sensitivity to fear in other rats, they may spend more time around photos with secure expressions. This study suggests that this tendency is a function of the transmission of facial expressions, as instinct dictates that rats adapt in response to signals from other individuals.

Data on the amount of activity during experience manipulation indirectly explained the increased sensitivity of the PWO group to pain. In the pain-experience manipulation, the PWO and PA groups received pain stimuli of similar intensity. However, during the experiment, the activity of the PWO group was lower than that of the PA group, particularly during the in-beep sound phase. It should be noted here that the reduced activity of animals in the PWO group may have been due to “freezing behavior”, as well as some sort of interaction with other conspecifics in order to receive the information. This finding may be related to the differences in experience between the PWO and PA groups due to the presence of social stimuli. In addition, compared to the pre-test, only the frequency of movement in the apparatus of the PWO group was reduced in the post-test. However, a change in state anxiety was transient during the experimental manipulation, whereas no clear difference was observed between the groups in the open-field examination one day after the post-test. The decrease in activity during the post-test in the PWO group may be due to the perceived danger from painful facial photographs rather than a temporary increase in anxiety caused by prior foot shocks.

It should be noted here that the previous study showed that foot shock affected the locomotor activity of rats in the open-field test later [26], whereas we found no significant differences between the PWO, PA and control groups. The reason for the difference in results between the previous study and the present study may be the interval between the administration of foot shock and the measurement of locomotor activity. In the previous study, the locomotor activity was measured immediately after the administration of foot shock, whereas in the present study, it was measured two days after the administration of foot shock. In addition, the intensity of foot shock in the previous study (1.0 mA) was higher than that in the present study (0.7 mA). These differences may explain the different results.

A study of human subjects showed that self-reported pain intensity modulates attention to the pain expressed by others [18]. This study revealed that human participants who experienced high pain intensity focused longer on a picture of a neutral expression than those who experienced low pain intensity. However, in the present study, the length of stay per entry of rats in the PA group remained the same between picture locations after manipulation. As such, prior pain experience was insufficient to change the rats’ sensitivity to the emotional expression of other conspecifics. This finding may indicate that rats experience difficulty associating physical senses with subsequent emotional signals from other conspecifics. In fact, the results of the present study are likely derived from the actual emotional signal of “real” rats, but more from the stimulus of the photographs. However, focusing on the functional role of facial expressions remains important because it enables comparison between rodent and human studies. In addition to previous studies showing that rats exhibit facial expressions [8] and can discriminate between facial expressions [12], the present study illustrates that facial expressions function socially in response to context. Thus, this study suggests that the facial expressions of rats have functional similarities comparable to those of humans. However, the extent to which humans and rats have similar functions requires further investigation.

The reason why the PA group no longer experienced a difference in the length of stay between the pain and neutral stimuli in the post-test after manipulation remains unclear. One possibility is that the PA group may have displayed certain prosocial behaviors in response to pain photos after manipulation. However, this situation may be ambivalent, because the pain photographs were typically avoided. In recent years, rats have been shown to exhibit prosocial behaviors, such as helping other conspecifics who are confined [27] or soaked [28]. In addition, studies of emotional contagion have demonstrated that rats subsequently engage in behavior that comforts individuals who suffer pain when witnessing foot shocks from other individuals [29]. However, increased sensitivity to pain stimuli in the PWO group may not have led to such prosocial behaviors. This possibility is consistent with other human studies, where the level of anxiety influenced the participants’ attention to pain expression [21]. Depending on the degree of fear or anxiety previously experienced, opting for avoidance behaviors rather than helping behaviors may be adaptive in rats.

It should be noted that there is another possibility that the painful experience of the animals in the PA group was attenuated during manipulation by some sort of interaction with the cagemate (i.e., caring behavior) after returning to the home cage because the animals were pair-housed throughout the period of acclimation and the experiment. The phenomenon of social buffering in rats is well known, wherein the interaction with other individuals of the same species after negative events, such as fear, reduces fear [30,31]. Therefore, sensitivity to the emotional signal of pain in animals in the PA group may have been suppressed by the attenuation of the painful experience through interaction with the cagemate. However, the interaction between animals during acclimation was outside the scope of this study; therefore, we had no objective data to explain this possibility. Future studies should investigate the possibility that the attenuation of painful experiences by interaction with cagemates reduces the sensitivity of emotional signals in animals.

Finally, this study may have potentially important implications for the study of emotional contagion [32,33,34], although this aspect was outside the focus of this study. First, it proposes a new aspect that focuses on the subsequent behavior of the injured side of the animal in terms of emotional contagion (i.e., corresponding to a demonstrator in a study on emotional contagion). Typically, in the field of emotional contagion related to pain, researchers have focused mainly on the emotional states or cognitive processes of observers when observing other individuals experiencing pain. Consequently, the current study illustrated that animals in the PWO group displayed decreased activity during manipulation, as well as a decreased frequency of movement in the post-test. This study, therefore, provides an opportunity to highlight the need for further research in the field of emotional contagion, especially for the emotional or cognitive process of an individual experiencing pain. Second, the present study is significant because it demonstrates the importance of vision via the use of photographs of emotional expression of other conspecifics as experimental stimuli, with strict control of the extraneous variables such as the odor or phonetical information. Langford et al. demonstrated that visual information is crucial for emotional contagion in mice [35]. Thus, the results of this study strengthen the argument that rodents use visual information during emotional signal processing.

However, this study has limitations in the context of emotional contagion. We did not set up a control condition in which the subject animal did not receive foot shock in the company of another individual. The reason for this was that this study focused on examining the effects of pain experiences on facial expression recognition. Pain is known to have several effects on the emotional contagion process: effects (a) of expressing pain; (b) effects on individuals who are not in pain but have observed pain in other individuals; and (c) effects of being observed while expressing pain. The literature on emotional contagion indicates that the influence of pain expression and observation is interconnected [22]. Therefore, in the future, additional research should be conducted using several of the aforementioned controls, which should also be considered necessary in terms of determining the influence of social context on the emotional contagion of pain.

## 5. Conclusions

According to a literature review, this study is the first to demonstrate that rats may change their behavior in response to emotional signals by referring to prior experience. In other words, this finding indicates that rodents may utilize the information they receive from other conspecifics to change their behavior in response to visual–emotional signals (i.e., emotional expression of pain from other conspecifics). Future research should examine the exact neurocognitive mechanisms to determine the effect of the socioemotional context and individual differences on anxiety in terms of the sensitivity of rats to emotional expression.

## Figures and Tables

**Figure 1 animals-14-01280-f001:**
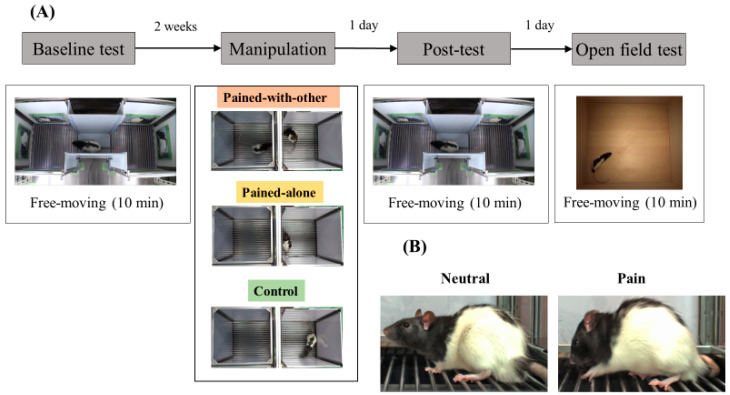
Flow of the procedure and image stimuli used in the preference test. (**A**) The experiment primarily comprised four phases, namely the baseline preference test, pain experience manipulation, the post-manipulation preference test, and the open-field test. For the manipulation of context, rats were assigned to three groups, namely pained-alone (PA), pained-with-other (PWO), and control. Animals in the PA group were placed alone in the shock room with a grid floor and received an electrical shock. Animals in the PWO group were placed in the shock room and electrically shocked similar to those in the PA group, while another rat was concurrently placed in the observation room. Animals in the control group were placed in the shock room but never received an electrical shock. In the baseline and post-manipulation test phases, each rat was placed in the center zone and left to explore the apparatus for 10 min. Finally, the open-field test was conducted to measure state anxiety. (**B**) Photographs of neutral expressions were taken after acclimating rats to the environment in the absence of painful stimuli. The photographs of pained expressions were taken as the rats received shocks to their feet.

**Figure 2 animals-14-01280-f002:**
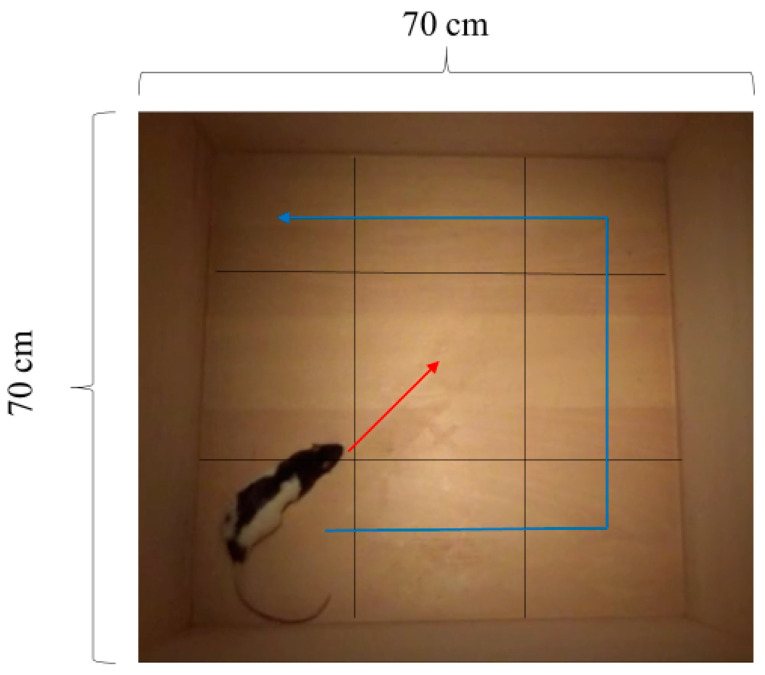
Illustration of the analytical method of outcomes for the open-field test. An entire field was manually divided into nine small blocks. The first index of the anxiety level was how many animals crossed the line of each block, which was calculated as the frequency of movement (blue line). The second index of the anxiety level was how many animals entered the central block of the field by crossing lines from other blocks, counted as entries into the central location (red line).

**Figure 3 animals-14-01280-f003:**
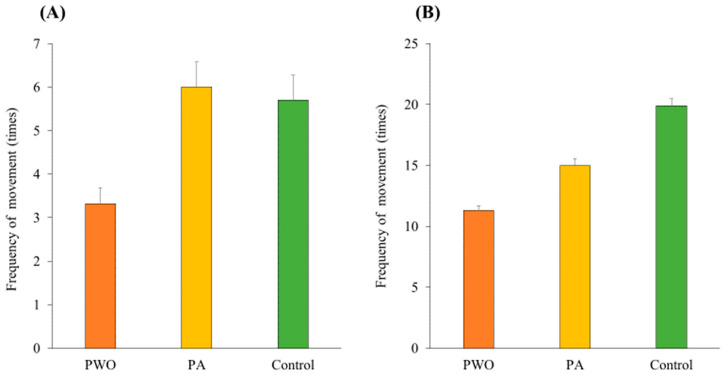
(**A**) Mean proportions of the frequency of movement in terms of level of in-beep phase activity during the manipulation. (**B**) Mean proportions of the frequency of movement in terms of level of activity in silent phase during manipulation. Orange panels represent the PWO group, yellow panels represent the PA group, and green panels represent the control group. Error bars represent the standard errors of the means.

**Figure 4 animals-14-01280-f004:**
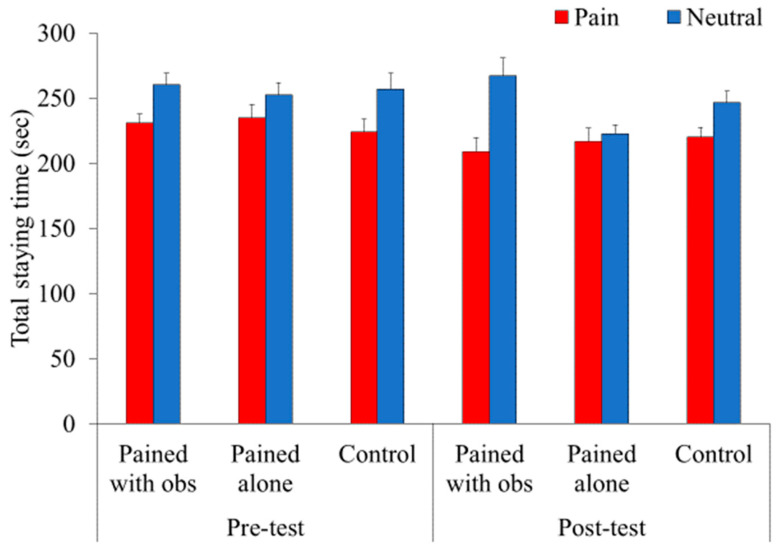
Mean proportion of total length of stay in each box (total lengths of stay are presented as a function of pre-/post-pain, context, and emotional expression). The index shows how long animals in each group stayed in either box with photographs of pain (red panel) or neutral dispositions (blue panel) independent of the number of entries into each box. Error bars represent the standard errors of the means.

**Figure 5 animals-14-01280-f005:**
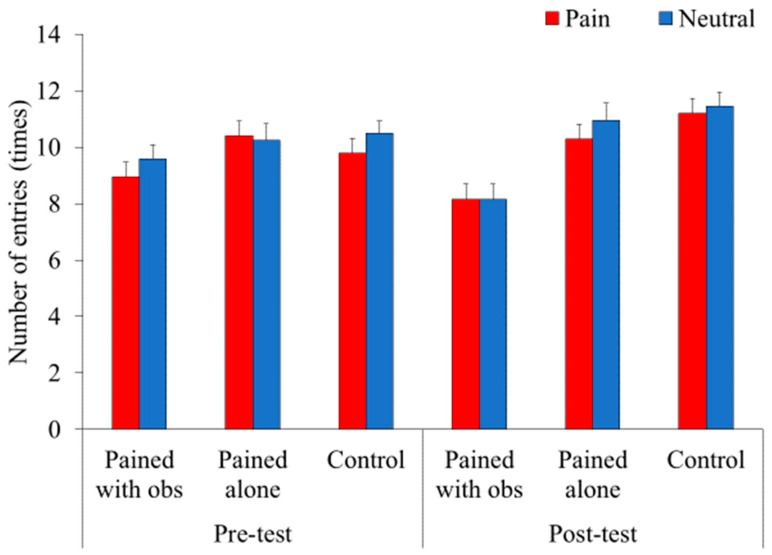
Mean proportion of the frequency of entry into each box. The index shows how many times animals in each group entered either box attached with photographs of pain (red panel) or neutral disposition (blue panel). Error bars represent the standard errors of the means.

**Figure 6 animals-14-01280-f006:**
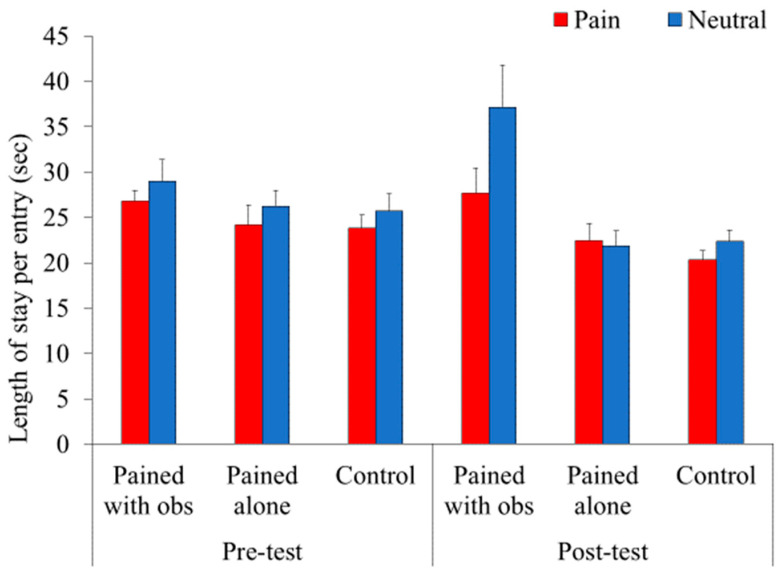
Mean proportion of length of stay per entry for each condition. The index was calculated by dividing the average total length of stay in each box by the frequency of entries into each box. Therefore, the index considers the difference of activity in the test box between each condition. Red panels represent the length of stay in the box with photographs depicting pain, and blue panels represent the length of stay in the box with photographs depicting a neutral disposition. Error bars represent the standard errors of the means.

**Figure 7 animals-14-01280-f007:**
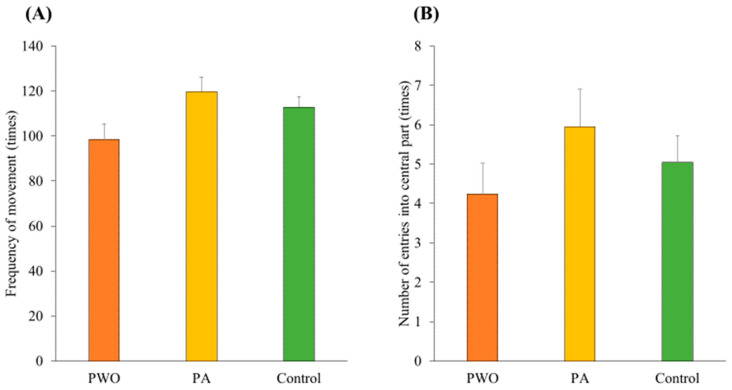
(**A**) Mean proportion of frequency of movement among the nine regions in the open field. (**B**) Mean proportion of frequency of entries into the central region of the open field. Orange panels represent the PWO group, yellow panels represent the PA group, and green panels represent the control group. Error bars represent the standard errors of the means.

## Data Availability

A raw dataset is available from the Open Science Framework (https://osf.io/5fhjr/?view_only=b258f7082f054124af3d2cca1b71a129) accessed on 10 September 2020.

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
