# Peer review of "Painful Experiences in Social Contexts Facilitate Sensitivity to Emotional Signals of Pain from Conspecifics in Laboratory Rats"

_animals, 2024, doi:10.3390/ani14091280_

Round 1
Reviewer 1 Report
Comments and Suggestions for Authors
In reviewing this paper, I have identified several areas that could be improved for clarity and coherence:
1. Writing: The manuscript would benefit from overall better writing to enhance readability and clarity.
2. Figure Panels: There is a need to visually highlight the differences between figure panels to aid in comparison and interpretation.
3. Figure Legends: Adding detailed information to figure legends can help readers understand the distinctions between the panels more effectively.
4. Key Findings: The manuscript should include a discussion of the only significant effect observed, which is the increased time per entry in the POW group. It would be helpful to explore potential explanations for why other differences were not detected.
5. Main Observation Interpretation: Consider discussing whether the decrease in activity in the POW group during manipulation can be attributed solely to the fact that they spend more time 'immobile' while interacting with other animals.
Additionally, I would like to point out that the conclusions drawn are not aligned with the presented results and appear overstated. It is crucial to ensure that the conclusions accurately reflect the findings to maintain the integrity of the manuscript.
Comments on the Quality of English LanguageNeeds revision
Author Response
We greatly appreciate your insightful comments. Please find our response to all the comments below.
Q1. Writing: The manuscript would benefit from overall better writing to enhance readability and clarity.
A1. In accordance with your comment, we have sent our manuscript for English proofreading; therefore, we believe that the manuscript will be more readable and clearer.
Q2. Figure Panels: There is a need to visually highlight the differences between figure panels to aid in comparison and interpretation.
A2. In accordance with your comment, we have modified all the figure panels to be colorful and distinguishable for easy comparison and interpretation of the data.
Q3. Figure Legends: Adding detailed information to figure legends can help readers understand the distinctions between the panels more effectively.
A3. In accordance with your comment, we added detailed information to the figure legends to improve understanding.
Q4. Key Findings: The manuscript should include a discussion of the only significant effect observed, which is the increased time per entry in the POW group. It would be helpful to explore potential explanations for why other differences were not detected.
A4. Thank you for pointing this out. As Reviewer 2 noted, it might be possible that the interaction with the cagemate after manipulation attenuates painful experiences of the animals in PA, so their sensitivity to the photographs of the pain of conspecifics was not observed in the study. We have added a description of this possibility in the Discussion section (p.12 line 411-422).
Q5. Main Observation Interpretation: Consider discussing whether the decrease in activity in the POW group during manipulation can be attributed solely to the fact that they spend more time 'immobile' while interacting with other animals.
A5. Thank you for your insightful comment. As you indicated, the reason why the activity in the PWO group during manipulation could not be attributed solely to being immobile while interacting with other animals. We consider that the index of the frequency of movement of animals in the PWO group during manipulation includes two different kinds of animal behavior: actual immobility (freezing) and interaction with other conspecifics. We have added a description of this possibility in the Discussion section (p.11 line 368-370), although it is difficult to differentiate these behaviors objectively from each other in our data.
Q6. Additionally, I would like to point out that the conclusions drawn are not aligned with the presented results and appear overstated. It is crucial to ensure that the conclusions accurately reflect the findings to maintain the integrity of the manuscript.
A6. Thank you for your comments. As you indicated, the descriptions in the conclusion included some expressions that were incongruent with the actual results; therefore, we modified them to be congruent with the results (p.13 line 450-454).
Reviewer 2 Report
Comments and Suggestions for Authors
In this article, the authors present an original experiment designed to determine whether, in laboratory rats, a previous painful experience influences the sensitivity of pain recognition in a conspecific.
The experiment is designed to carry out an initial measurement of the animals' preference towards two compartments, one comprising photos of conspecifics in a neutral attitude, the second comprising photos of conspecifics in a painful posture (arched back). The animals are then randomly divided into three groups, one receiving painful stimulations alone (PA), the second receiving painful stimulations in the presence of a conspecific (PWO), and the third receiving no stimulation at all (Control). The animals' preference for the two compartments is then tested again before a final test in an open field to quantify anxiety levels. Behavioural measurements are essentially based on the mobility of individuals in the different compartments, and parametric tests are used to compare the different groups and situations.
The results show that during the painful experience, individuals stimulated in the presence of a conspecific were less mobile than those stimulated alone. According to the authors, this is indirect evidence of an increased sensitivity in the group stimulated in the presence of a conspecific. The intensity of stimulation is in fact identical for both groups of animals, so that the presence of a conspecific may reasonably explain an increased response in terms of freezing.
After the painful experience, the authors show a marked difference between the groups, with an increased preference for the safe (i.e. non-painful) compartment for animals that had been stimulated in the presence of a conspecific. The results therefore demonstrate the ability of rats to interpret the facial expressions of their mates, and to take account of their own experience to adjust their behavioural response.
Comments/Questions on the study design
Line 173 states that animals were randomly assigned to one of the three conditions, but the method used is not specified. Please indicate the method used to randomly allocate animals in the experimental groups.
Since animals were housed by pairs of individuals, one may wonder if interactions between pairs of animals might contribute to the attenuation of the painful experience. Did you notice anything once the animals were placed back in their home cage after the pain experience manipulation?
In the pain-with-other conditions, a rat was placed in the non-stimulated compartment. Which animal was used for this? Is it an animal from the same experimental cohort, is it the cage mate, or a completely naive animal?
Blinding is a strategy used to minimise subjective biases such as knowledge of treatment applied to an individual. Please indicate whether blinding was used at the different stages of the experiment (allocation, conduct of the experiment, outcome assessment, and data analysis).
Please clearly indicate the main outcome measure in the open-field test. It might be interesting to distinguish movements in and out of the central region from movements in and out of the periphery of the box.
Statistical analyses: please indicate whether the statistical tests used have taken into account the pairing of measurements.
Comments/Questions on the Results section
Careful observation of the data table suggests paying a particular attention to rat#14 belonging to the PWO group, and to a lesser extent to rat with ID#1. Compared to the other individuals of this group, rat #14 seems to have spent much more time in the neutral compartment after the painful experience (x4.2). I wouldn't be surprised if this rat showed less mobility in the open-field test and therefore a greater level of anxiety. If this animal appears to be a true outlier, then the average proportion of length of stay (and its standard error) might be substantially biased.
Discussion
Overall, the results are quite convincing. The discussion is well conducted and covers all the key points of the study. It is also important to note that, in addition to a painful posture, rats are also capable of emitting other signals, particularly ultrasound, which contribute significantly to the sharing and transmission of experience. In return, they can also benefit from empathic behaviours that are increasingly reported in the literature.
It remains to be elucidated why animals stimulated alone do not show a marked preference for the neutral compartment compared to those stimulated in the presence of a conspecific. Although the emotional contagion is not the focus in this study, this article highlights the importance of addressing this notion in future studies.
Comments on the Quality of English LanguageAs far as I can tell, English language is alright
Author Response
Thank you for your insightful comments. Please find our response to your comments below.
Q1. Line 173 states that animals were randomly assigned to one of the three conditions, but the method used is not specified. Please indicate the method used to randomly allocate animals in the experimental groups.
A1. Thank you for pointing this out. Random numbers were created using Excel’s random function tied to each animal, and the sequential order of the animals was changed from the original ID based on the random number and allocated to each group. We have added this information to the Pain Experience Manipulation section (p. 5 line 193-196).
Q2. Since animals were housed by pairs of individuals, one may wonder if interactions between pairs of animals might contribute to the attenuation of the painful experience. Did you notice anything once the animals were placed back in their home cage after the pain experience manipulation?
A2. Thank you for your insightful comment. In fact, I noticed that the cagemate possibly approached and cared more for the animal that was foot-shocked when I returned them to the home cage. Of course, this is in my subjective sense, and we did not have this data as it was outside the scope of this study.. However, this may explain why the animals in the PA group did not show enhanced sensitivity to photographs of conspecific pain. In other words, it might be possible that the interaction with the cagemate after manipulation attenuates painful experiences of the animals in the PA group, so their sensitivity to the photographs of pain of the conspecifics was not observed in this study. We have added a description of this possibility in the Discussion section (p.12 line 411-422).
Q3. In the pain-with-other conditions, a rat was placed in the non-stimulated compartment. Which animal was used for this? Is it an animal from the same experimental cohort, is it the cage mate, or a completely naive animal?
A3. Thank you for pointing this out. The observer rats were 20 novel individuals that were not included in any conditions of this experiment and were not cage mates for the experimental animals in the PWO group. We have added this information to the Pain Experience Manipulation section (p. 5 line 209-211).
Q4. Blinding is a strategy used to minimise subjective biases such as knowledge of treatment applied to an individual. Please indicate whether blinding was used at the different stages of the experiment (allocation, conduct of the experiment, outcome assessment, and data analysis).
A4. Thank you for pointing this out. In fact, we were a small research group (only three people) when we conducted this study and had insufficient resources; therefore, we could not perform all processes in a blinded manner. We have noted the allocation of animals in each group was random. The experiment was not blinded because the first author conducted all the experimental procedures. However, the outcome assessment was conducted by the second author, who was not informed on any information regarding the experimental group; thus, this process was completely blinded. The process of data analysis was conducted by the first author, so the process was not blinded. We have added this information to the Procedure section (p.5 line 182-184) and its footnote.
Q5. Please clearly indicate the main outcome measure in the open-field test. It might be interesting to distinguish movements in and out of the central region from movements in and out of the periphery of the box.
A5. Thank you for pointing this out. In accordance with your comments, we calculated the number of entries into the central region independent of the frequency of movement between the nine regions. We have added a description explaining the number of entries into the central region as an outcome in the open-field section (p.6 line 220-225) and Figure 2 to explain how we created and counted each outcome. We also added a description of the number of entries into the central region in the open-field section (p.10 line 343-346) and Figure 7 (B).
Q6. Statistical analyses: please indicate whether the statistical tests used have taken into account the pairing of measurements.
A6. Thank you for pointing this out. In accordance with your comments, we firstly added the description for the experimental design of this study to clarify that each factor was either between the subject factor or within the subject factor in the “Design” section (p.3 line 131-135). Furthermore, we have added a description of either “mixed factorial design” or “between subjects” for each ANOVA analysis in the Results section (p.6 – p10).
Q7. Careful observation of the data table suggests paying a particular attention to rat#14 belonging to the PWO group, and to a lesser extent to rat with ID#1. Compared to the other individuals of this group, rat #14 seems to have spent much more time in the neutral compartment after the painful experience (x4.2). I wouldn't be surprised if this rat showed less mobility in the open-field test and therefore a greater level of anxiety. If this animal appears to be a true outlier, then the average proportion of length of stay (and its standard error) might be substantially biased.
A7. Thank you for your insightful comment. As you have indicated, rats #1 and #14 in the PWO group spent a lot of time in the neutral compartment after the painful experience. However, the animals also spent more time in the neutral compartment even before the painful experience, and the frequency of movement between boxes was not significantly lower than that in other animals. Therefore, we consider it difficult to regard these animals as true outliers based solely on the data. It should be noted that we could not obtain data for rats #14 and #15 in the open-field test because of missing recordings. We apologize for the erroneous omission of this information in our manuscript. We have added a description of this information in the open-field section (p.10 line 336-337).
Q8. It remains to be elucidated why animals stimulated alone do not show a marked preference for the neutral compartment compared to those stimulated in the presence of a conspecific. Although the emotional contagion is not the focus in this study, this article highlights the importance of addressing this notion in future studies.
A8. Thank you for your comments. As noted above, it might be possible that the interaction with the cagemate after manipulation attenuates painful experiences of the animals in the PA group, so their sensitivity to photographs depicting pain of conspecifics was not observed in this study. We have added a description of this possibility in the Discussion section (p.12 line 411-422).
Reviewer 3 Report
Comments and Suggestions for Authors
The presented manuscript addresses a very interesting experimental, philosophical, and ethical issue related to pain research. The difference between nociception and pain is well known, but relatively little is known about pain-related emotions in experimental animals. It is important to note that rodents, especially nocturnal ones, do not rely as much on visual stimuli as on auditory and olfactory ones. In this connection, some questions arise regarding the execution of the experiment and the discussion of the results.
It is important to specify which animals participated in the experimental pairs. Since the rats were housed in pairs in a home cage, it should be specified whether the same pairs or animals that had not been in contact until the time of the experiment were used in the test.
The role of the sound stimulus (beep) is not adequately discussed because it modifies experiments in conditioning learning.
In Fig.3 and 5, perhaps the time should be expressed in seconds.
In my opinion, the experiment should include the same treatments and groups, but without visual stimuli of suffering or presence, to compare the influence of visual stimuli themselves.
When studying such sensitive parameters of behavior, which include a variety of interactions between the tested animals, it is necessary to specify: in which room the study is conducted, and where the already tested animals from the individual groups are located. The time of the experiment should also be specified because this would reflect their inherent circadian activity.
Such omissions can lead to ambiguous results or summation of the influence of stimulation.
Author Response
Thank you for your insightful comments. Please find our response to your comments below:
Q1. It is important to specify which animals participated in the experimental pairs. Since the rats were housed in pairs in a home cage, it should be specified whether the same pairs or animals that had not been in contact until the time of the experiment were used in the test.
A1. Thank you for pointing this out. Are we correct in considering that "animals participated in experimental pairs" refers to the experimental animal and an observer rat in the PWO group? The observer rats were 20 novel individuals that were not included in any groups of this experiment and were not cage mates for the experimental animals in the PWO group. We have added this information to the Pain Experience Manipulation section (p. 5 line 209-212).
Q2. The role of the sound stimulus (beep) is not adequately discussed because it modifies experiments in conditioning learning.
A2. Thank you for pointing this out. The beeping sound was used as a conditioned stimulus to create a strong fear association between the electrical shock and the beeping sound, rather than making animals have an association between the electric shock and other environmental situations, and/or making animals have a vague sense of fear in the environment. We have added this description to the Pain Experience Manipulation section (p. 5 line 202-205).
Q3. In Fig.3 and 5, perhaps the time should be expressed in seconds.
A3. Thank you for pointing this out. We have modified this.
Q4. In my opinion, the experiment should include the same treatments and groups, but without visual stimuli of suffering or presence, to compare the influence of visual stimuli themselves.
A4. Thank you for your insightful comment. As you indicated, including conditions without visual stimuli to photographs depicting pain or neutral dispositions could be useful for comparing the influence of visual stimuli. We have previously considered this as well. Thus, in our previous study, we examined the behavior of rats that were modified by non-meaningful visual stimuli, such as mosaic or random patterns (Nakashima et al., 2015). The results of this study indicated that unmeaningful visual stimuli, such as mosaic or random patterns, did not influence the behavior of animals (i.e., animals could not differentiate between a pair of this type of stimuli). Therefore, we believe that the presence of the visual stimuli alone did not influence the results of this study. In addition, if we incorporated such conditions, the sample size would increase twofold and might negatively affect our study from the ethical perspective of "Reduction”; therefore, we did not include such conditions in the study.
Q5. When studying such sensitive parameters of behavior, which include a variety of interactions between the tested animals, it is necessary to specify: in which room the study is conducted, and where the already tested animals from the individual groups are located. The time of the experiment should also be specified because this would reflect their inherent circadian activity. Such omissions can lead to ambiguous results or summation of the influence of stimulation.
A5. Thank you for your comments. The rooms used for breeding, pain experience manipulation, and the baseline/post-test were separated and shielded from each other. Each experimental procedure was conducted between 2:00 pm and 6:00 pm, when the activity of the animals was high in the rearing system. We have added this information to the Procedure section (p. 5 line 184-187).
Round 2
Reviewer 1 Report
Comments and Suggestions for Authors
The manuscript benefited substantially form the Reviewers' comments. The manuscript's readability also improved.
I acknowledge the effort taken by the Authors in addressing my prior points.
I still think that the manuscript will further benefit from improving the readability of figure legends. For example the authors write:
'Figure 7. The anxiety level during the open-field test. The left panel represents the mean proportion of frequency of movement among the nine regions in the open field (A). The right panel ...'
It should state:
'Figure 7. (A) mean proportion of frequency of movement among the nine regions in the open field . (B)...'
Mentioning 'right/left panel' together with (A/B...) is unnecessary.
This applies too to Figure 3.
Additionally:
In Figure 7 legend the Authors stated:
'proportion of frequency of movement'. However, the y axes of panel A suggests total time(?). Please clarify.
Author Response
We greatly appreciate your insightful comments. Please find our response to all the comments below.
Q1. I still think that the manuscript will further benefit from improving the readability of figure legends. For example the authors write:
'Figure 7. The anxiety level during the open-field test. The left panel represents the mean proportion of frequency of movement among the nine regions in the open field (A). The right panel ...'
It should state:
'Figure 7. (A) mean proportion of frequency of movement among the nine regions in the open field . (B)...'
Mentioning 'right/left panel' together with (A/B...) is unnecessary.
This applies too to Figure 3.
A1. In accordance with your comment, we have modified these figure legends.
Q2. Additionally:
In Figure 7 legend the Authors stated:
'proportion of frequency of movement'. However, the y axes of panel A suggests total time(?). Please clarify.
A2. Thank you for pointing this out. In Figure 7, we actually counted the proportion of frequency of movement among the nine regions. Therefore, the description of the y-axes in panel A is correct.
Reviewer 3 Report
Comments and Suggestions for Authors
Despite the seemingly large corrections in the text, it is clear that most are edits related to the English language, not scientific results. The most corrections are non-significant changes in the color of the figures, but there are no symbols for a significant difference between the specific groups from the post hoc analysis. The reader should guess from the means and SEM and implement the factor analysis to understand which groups are significantly different. Differences in the control groups are not discussed at all, for example, the reduced stay in the room with a picture of a suffering animal before and after the test. Such differences were not hypothesized because controls did not witness the shock.
Are the controls subjected to sound in the same manner as the experimental groups?
The results of this study are different from the previous data in the literature and need a more precise presentation and detailed discussion in light of already published results.
In Figure 7, there were no significant differences in the locomotor activity, including in the central area, of the rats after they were subjected to shock compared to the controls. How would the authors discuss this result given that this stressor is a model for inducing anxiety behavior? (Rida Nisar, Zehra Batool, Saida Haider, Electric foot-shock induces neurobehavioral aberrations due to imbalance in oxidative status, stress hormone, neurochemical profile, and irregular cortical-beta wave pattern in rats: A validated animal model of anxiety, Life Sciences, Volume 323, 2023, 121707, ISSN 0024-3205).
The statement "Therefore, sensitivity to the emotional signal of pain in animals in the PA group may have been suppressed by attenuation of the painful experience through interaction with the cage-mate." can be referred to the group of rats that witnessed the painful shock.
Author Response
We greatly appreciate your insightful comments. Please find our response to all the comments below.
Q1. Despite the seemingly large corrections in the text, it is clear that most are edits related to the English language, not scientific results. The most corrections are non-significant changes in the color of the figures, but there are no symbols for a significant difference between the specific groups from the post hoc analysis. The reader should guess from the means and SEM and implement the factor analysis to understand which groups are significantly different. Differences in the control groups are not discussed at all, for example, the reduced stay in the room with a picture of a suffering animal before and after the test. Such differences were not hypothesized because controls did not witness the shock.
A1. Thank you very much for your insightful comments. In accordance with your comments, we have added the details of the post hoc analysis in some of the data in the results section. Although you have noted the point that there is significant difference between the length of stay in the room with photographs of pain at baseline and that at post-test in the control group, we did not find any significant differences between them. As we noted in the Length of stay per entry section (p.9 line 315-318), 'Subsequent analyses revealed that the pre/post-pain × emotional expression interaction was significant in the PWO condition (F (1, 56) = 6.58, p < 0.05, ηp2 = 0.11), but not in the PA (F (1, 56) = 0.81, p = 0.37, ηp2 = 0.01) or control (F (1, 56) = 0.003, p = 0.96, ηp2 < 0.01) condition'. This means that there was no significant interaction between pre/post-pain and emotional expression in the control groups. Therefore, we believe that we should not perform a post hoc analysis for the control group and should not discuss it further.
Q2. Are the controls subjected to sound in the same manner as the experimental groups?
A2. Thank you for pointing this out. We made animals in control group experienced beeping sound in the same manner as the experimental groups. We so that added this in Pain Experience Manipulation section (p.5 line 211).
Q3. The results of this study are different from the previous data in the literature and need a more precise presentation and detailed discussion in light of already published results.
In Figure 7, there were no significant differences in the locomotor activity, including in the central area, of the rats after they were subjected to shock compared to the controls. How would the authors discuss this result given that this stressor is a model for inducing anxiety behavior? (Rida Nisar, Zehra Batool, Saida Haider, Electric foot-shock induces neurobehavioral aberrations due to imbalance in oxidative status, stress hormone, neurochemical profile, and irregular cortical-beta wave pattern in rats: A validated animal model of anxiety, Life Sciences, Volume 323, 2023, 121707, ISSN 0024-3205).
The statement "Therefore, sensitivity to the emotional signal of pain in animals in the PA group may have been suppressed by attenuation of the painful experience through interaction with the cage-mate." can be referred to the group of rats that witnessed the painful shock.
A3. Thank you for your insightful comment and for suggesting an earlier study. We believe that the difference between our study and the study you mention is due to the differences in the procedures in our study and the study you mention. The most important difference is the timing of the locomotor activity measurements. In the previous study, the locomotor activity of the animals in the open field test was measured immediately after the administration of the foot shocks, whereas in the present study it was measured two days after the administration of the foot shocks. In addition, the intensity of the foot shocks is also different between the present study (0.7 mA) and the previous study (1.0 mA). We have added a description of the reason for the difference between the present study and the previous study in the Discussion section (p.11 line 380).